# Primary Ciliary Dyskinesia in a Portuguese Bronchiectasis Outpatient Clinic

**DOI:** 10.3390/genes14030541

**Published:** 2023-02-21

**Authors:** Eduarda Milheiro Tinoco, Ana Rita Gigante, Edite Ferreira, Inês Sanches, Rute Pereira, Rosália Sá, Regina Monteiro, Mário Sousa, Ivone Pascoal

**Affiliations:** 1Department of Pulmonology, Centro Hospitalar de Vila Nova de Gaia/Espinho (CHVNG/E), 4434-502 Vila Nova de Gaia, Portugal; 2UMIB-Unit for Multidisciplinary Research in Biomedicine, ITR-Laboratory for Integrative and Translational Research in Population Health, University of Porto, 4050-313 Porto, Portugal; 3Department of Otorhinolaryngology, Centro Hospitalar de Vila Nova de Gaia/Espinho (CHVNG/E), 4434-502 Vila Nova de Gaia, Portugal; 4Laboratory of Cell Biology, Department of Microscopy, ICBAS-School of Medicine and Biomedical Sciences, 4050-313 Porto, Portugal

**Keywords:** primary ciliary dyskinesia, ciliary function, transmission electron microscopy, next-generation sequencing, genetics

## Abstract

Primary ciliary dyskinesia (PCD) is a rare hereditary condition characterized by decreased mucociliary clearance of the airways and a compromised reproductive system, resulting in male and female infertility. Several mutations with varied clinical and pathological features have been documented, making diagnosis a challenging process. The purpose of this study is to describe the clinical and pathological features of Portuguese patients with PCD and to examine their genetic variants. A retrospective observational analysis was conducted with patients who were being monitored at a bronchiectasis outpatient clinic in 2022 and had a confirmed or high-likelihood diagnosis of PCD. In total, 17 patients were included in the study, with 12 (66.7%) having PCD confirmed and 5 (29.4%) having a high-likelihood diagnosis. Furthermore, 12 patients were subjected to transmission electron microscopy (TEM), with 7 (58.3%) exhibiting one hallmark defect. Genetic test data was obtained for all 17 patients, with 7 of them (41.2%) displaying a pathogenic/likely pathogenic mutation in homozygosity. To summarize, PCD is an uncommon but significant hereditary illness with consequences regarding morbidity and mortality. Despite the lack of a specific treatment, it is critical to confirm the diagnosis with genetic testing in order to effectively manage the disease and its accompanying disorders.

## 1. Introduction

Primary ciliary dyskinesia (PCD) is a rare genetic disease that is primarily inherited as an autosomal recessive trait. It belongs to a clinically and genetically diverse group of respiratory ciliopathies characterized by decreased mucociliary clearance of the airways, recurrent infections, and bronchiectasis [1]. Furthermore, ciliary abnormalities can impair the reproductive system, resulting in male and female infertility; male infertility due to immotile or dysmotile spermatozoa and female subfertility due to the lack of ciliary motions in the fallopian tubes [2]. Although the prevalence of PCD varies widely among populations, it is predicted to afflict 1:2000 to 1:20,000 Europeans [3,4].

Cilia are responsible for various physiologic functions, including effective mucociliary clearance. The axoneme, or core of a cilium or flagellum, is an organized structure of microtubules with a 9 + 2 or 9 + 0 configuration, classified as motile or non-motile. Motile cilia with a 9 + 2 arrangement are found predominately in the upper and lower respiratory tracts, as well as in the female reproductive tract, whereas the flagellum is required for spermatozoa motility. A normal cilium has a circumference of nine microtubule doublets, with another pair in the center (central apparatus or central complex). Radial spoke proteins link the outer doublets to the center complex, and dynein and nexin proteins anchor them to one another (Figure 1 and Figure 2) [5].

PCD diagnosis is complex and requires a skilled team of clinicians, scientists, and microscopists. When confronted with diverse PCD symptoms, physicians can resort to the PICADAR score to aid in the appropriate referral for diagnostic testing. The PICADAR score is a seven-point questionnaire-based prediction tool that assists in determining the likelihood of PCD [1,6]. The score spans from 0 to 14, with a sensitivity and specificity of 0.90 and 0.17 for > 5 points, respectively [1]. 

According to the European Consensus Statement, there is no single gold standard diagnostic test for PCD. Consequently, the clinical diagnosis must be confirmed by a combination of technically demanding tests, such as nasal nitric oxide (nNO), high-speed video microscopy (HSVM), transmission electron microscopy (TEM), and genetic testing using, for example, next-generation sequencing (NGS) or Sanger sequencing [1]. 

The present study sought to describe the clinical and pathological features of Portuguese patients with PCD and to examine their genetic variants.

## 2. Materials and Methods

### 2.1. Ethical Considerations

All ethical guidelines were followed, with clinical data and biological material obtained under strict individual anonymity. This work did not involve human or animal experiments and thus the provisions of the Declaration of Helsinki, as revised in Tokyo 2004, do not apply to this work. Clinical files were reviewed under strict individual confidentiality for demographic, clinical, functional, radiological, and microbiological data, as well as diagnostic exams. The protocol for the study was approved by the Vila Nova de Gaia/Espinho Hospital Center (CHVNG/E) ethics committee. Biological material from the patients was obtained after written informed consent and used in experiments, according to the Joint Ethics Committee of the Hospital and University, CHUP/ICBAS approval number 2020-094 (077-DEFI-078-CE).

### 2.2. Patient Data

During 2022 (January–August), a retrospective observational single-center study was conducted in patients with a confirmed or highly-likely PCD diagnosis, followed in an adult bronchiectasis outpatient clinic at CHVNG/E.

A confirmed diagnosis was defined when a patient presented a clinical diagnosis of PCD (assumed with suggestive symptoms including persistent wet cough, *situs* anomalies, congenital heart defects, persistent rhinitis, chronic middle ear disease with or without hearing loss, a history in term infants of neonatal upper and lower respiratory symptoms, or admissions to neonatal invasive care), plus a hallmark ultrastructural ciliary defect [absence of outer dynein arm (ODA), combined absence of ODA and inner dynein arm (IDA), or absence of IDA combined with microtubular disarrangement (Figure 3 and Figure 4)], or an unambiguous biallelic mutation in a PCD-causing gene [1].

A high-likelihood diagnosis was assumed when at least two of the four main clinical features of PCD were present: unexplained neonatal respiratory distress (NRD) in a full-term infant, daily year-round cough starting before six months of age, daily year-round nasal congestion beginning before six months of age, or organ laterality defect. After excluding cystic fibrosis, these were paired with low levels of nNO (<356 ppb) or low fractional exhaled nitric oxide (FeNO) (<25 ppb). Even if the previously described criteria were not completely met, cases with Kartagener’s syndrome (*situs inversus*, chronic sinusitis, and bronchiectasis) were termed PCD [3].

### 2.3. Sample Collection

The patient peripheral blood used for DNA extraction was collected in EDTA tubes (VACUETTE, Porto, Portugal).

The nasal cells were obtained by nasal brushing, using a cytology soft sterile brush (Endobrush, Biogyn SNC, Mirandola, Italy), in both nostrils. After brushing, the cells of one nostril were placed in Medium 199 (Gibco, Life Technologies/ ThermoFisher Scientific Inc, Waltham, Massachusetts USA), supplemented with 50 µg/mL PenStrep, for HSVM analysis. The cells from the other nostril were placed in fixative (2.5% glutaraldehyde in cacodylate buffer 0.1 M) for TEM analysis.

### 2.4. Transmission Electron Microscopy

Nasal samples were analyzed as previously reported [7]. Briefly, the samples were initially fixed with 2.5% glutaraldehyde (Sigma-Aldrich, St. Louis, MO, USA) in 0.1 M cacodylate buffer (Merck, Darmstadt, Germany), 7.2 pH, for 2 h at room temperature (RT); then, post-fixed with 2% osmium tetroxide (Merck) in a buffer for 2 h at 4 °C and dehydrated in a graded ethanol series (VWR, Radnor, PA, USA). Afterwards, the samples were treated with 1% tannic acid (Merck) in 100% ethanol and embedded in epoxy resin (Epon, Sigma-Aldrich). Semithin and ultrathin sections were cut on an LKB-ultramicrotome (Leica Microsystems, Wetzlar, Germany) using diamond knives (Diatome, Hatfield Twp, PA, USA). Suitable areas of ciliated cells were selected in semithin sections (1 µm) and stained with methylene blue-Azur II (Merck). Ultrathin sections were retrieved on copper grids (Taab, Berks, England). After double-contrasting with aqueous uranyl acetate (BDH, Poole, England) and lead citrate (Merck), they were observed and photographed on a JEOL 100CXII transmission electron microscope (JEOL, Tokyo, Japan), operated at 60 kV.

TEM analyses were performed at the Royal Brompton Hospital (London, UK) in 3 cases, and the others were performed at ICBAS-UP. In the cases performed at ICBAS-UP, TEM data are additionally presented as quantitative morphologic analyses and quantitative ciliary beat axis analyses.

### 2.5. Quantitative TEM Morphologic Analysis

The cilia axoneme ultrastructure was evaluated at high magnifications in transverse sections. The diagnosis was based on the presence of a systematic defect in any of the axonemal structures, [8] according to the quantitative methods provided by the international consensus guideline for reporting transmission electron microscopy results in the diagnosis of primary ciliary dyskinesia (BEAT PCD TEM Criteria) [9].

### 2.6. Quantitative TEM Ciliary Beat Axis Analysis

The ciliary beat axis and the ciliary deviation were evaluated with a minimum of 100 transverse sections examined after printing. In the printed images, a line was drawn parallel to the central microtubules. Based on the main orientation of the drawn lines, a perpendicular reference line was then chosen and the angle of each line parallel to the reference line was calculated and subtracted from the mean, with the differences being close to zero. The standard deviation (SD) of these differences corresponds to the ciliary beat axis and ciliary deviation [10].

### 2.7. Ciliary Beat Frequency and Beat Patterns by High-Speed Video-Microscopy

Ciliary beat frequency (CBF) and ciliary beat patterns (CBP) were evaluated as previously described, with some adaptations. [11] Briefly, ciliated cells that were placed in Medium 199 immediately upon arrival at the laboratory were placed at 37 °C until analysis. The tissue-culture room was pre-warmed to 27 °C. An analysis was conducted in an inverted microscope (Olympus, Nikon, Tokyo, Japan) and the cells examined using a ×40 interference contrast lens. Undisrupted ciliated clusters devoid of mucus were selected for study. Beating ciliated cells were recorded using a digital high-speed video camera (Semiconductor Vita 5000, Pixelink/Navitar, Inc., New York, NY, USA) at a rate of 300 to 400 frames per second. The camera allowed video sequences to be recorded and played back at reduced frame rates or frame by frame.

The CBF was analyzed by two methods, the manual method using ImageJ, version 1.53e [12], and by the semi-automated method with the CiliarMove program. [13] For the manual analysis, in all included groups of beating cilia, it registered the number of frames required to complete 10 cycles, which were then converted to CBF using the calculation (CBF=total number of frames/(number frames for 10 beats) × 10). In all cases, more than 10 ciliated clusters were analyzed, and, in each, 3 distinct regions were evaluated with the 2 methods.

To assess CBP, each edge was analyzed using Image J. A coordinated ciliary beat in a back-and-forth movement along the entire epithelial edge was defined as normal. The dyskinetic beat pattern was categorized into eight distinct CBPs by a modification of previously reported descriptions (immotile, circular, hyperkinetic, hyperkinetic with reduced beat amplitude, asynchronous, asynchronous with reduced beat amplitude, stiff, and synchronous with reduced beat amplitude) [14,15].

### 2.8. Nucleic Acids Extraction

Genomic DNA was extracted from peripheral blood leukocytes, following the salting-out method, [16] quantified by a NanoDrop spectrophotometer ND-1000 (Version 3.3; LifeTechnologies; CA, USA), and stored at 4 °C until further use.

### 2.9. Next-Generation Sequencing

A next-generation sequencing analysis was performed involving all exons and intron-exon transitions from a panel of 42 genes associated with primary ciliary dyskinesia: *CCDC103*, *CCDC39*, *CCDC40*, *CCDC65*, *CCNO*, *CENPF*, *CFAP 298*, *CFAP300*, *CFTR*, *DNAAF1*, *DNAAF2*, *DNAAF3*, *DNAAF4*, *DNAAF5*, *DNAAF6*, *DNAAF11*, *DNAH1*, *DNAH11*, *DNAH5*, *DNAH9*, *DNAI1*, *DNAI2*, *DNAL1*, *DRC1*, *GAS2L2*, *GAS8*, *HYDIN*, *INVS*, *LZTFL1*, *MCIDAS*, *NME8*, *ODAD1*, *ODAD2*, *OFD1*, *RPGR*, *RSPH1*, *RSPH3*, *RSPH4A*, *RSPH9*, *SPAG1*, *STK36*, and *ZMYND10.*

The libraries were prepared with the Twist Human Core Exome Plus kit. A new sequencing generation was carried out on an Illumina platform, with 99.46% of the bases having coverage greater than 20x. Sequence alignment was performed with the reference genome (GRCh37/hg19). Bioinformatics analyses were carried out with our own pipeline using different bioinformatics tools. The analysis of copy number variations (CNVs) was performed using our own bioinformatics pipeline. All pathogenic, probably pathogenic, potentially pathogenic variants and variants of uncertain clinical significance were reported. All pathogenic/probably pathogenic variants were confirmed by Sanger sequencing. The classification of variants follows the recommendations of the “American College of Medical Genetics and Genomics” (ACMG) [17]. The methodology used has an analytical sensitivity of 99% for the detection of variants, punctual and small insertions/deletions.

Eight of the genetic analyzes were performed at Synlab, Medical Genetics (Porto, Portugal) and eight at IPATIMUP/i3S (Instituto de Patologia e Imunologia Molecular da Universidade do Porto/ Instituto de Investigação e Inovação em Saúde), IPATIMUP Diagnostics, Unit of Medical Genetics, University of Porto (Porto, Portugal).

A few differences were observed between the two laboratories, which are not important for the present study. Nevertheless, at IPATIMUP, the libraries were prepared with the Ion AmpliSeq Exome Panel (Hi-Q) Kit (Ion Torrent, ThermoFisher Scientific Inc, Waltham, Massachusetts USA), and the new sequencing generation was performed in the Ion semiconductor sequencing platform, with 97.1% of the bases having coverage greater than 20x (minimum horizontal coverage of 20X: percentage of bases of the target genes read at least 20X), and an average vertical coverage of sequencing (average of readings of each base of the target genes of study) of 155X. Sequence alignment was performed with the reference genome UCSC hg19. The next-generation sequencing analysis was performed involving all exons and intron-exon transitions from a panel of 27 genes associated with primary ciliary dyskinesia, the majority were covered as above: *CCDC103*, *CCDC39*, *CCDC40*, *CCDC65*, *CCNO*, *CFAP298*, *DNAAF1*, *DNAAF2*, *DNAAF3*, *DNAAF4*, *DNAAF5*, *DNAAF11*, *DNAH5*, *DNAH11*, *DNAI1*, *DNAI2*, *DNAL1*, *DRC1*, *HYDIN*, *NME8*, *ODAD1*, *ODAD2*, *RSPH1*, *RSPH4A*, *RSPH9*, *SPAG1*, *ZMYND10.*

### 2.10. Statistical Analysis

The IBM Statistical Package for the Social Sciences (SPSS)^®^ was used for descriptive statistical analysis (version 28.0, Armonk, NY: IBM Corp). According to the data distribution, categorical variables were expressed as frequencies (n) and percentages (%), while continuous variables were characterized as mean ± standard deviation or median and interquartile range.

## 3. Results

The majority of the 17 patients were female (70.6%), with a mean age at diagnosis of 34.1 (±11.3) years and a mean PICADAR score (predicts the risk of developing PCD) of 6.2. (Table 1). Four patients had first-degree relatives with PCD, of which three were sisters included in the present study and the fourth sibling was followed at another hospital. The patients reported a history of newborn respiratory problems, *situs inversus*, chronic productive cough, and infertility in 47.1%, 17.6%, 94.1%, and 35.3% of cases, respectively.

The HSVM analysis demonstrated a lower mean ciliary beat frequency [CBF (N 7.0–19.0 Hrz): 6.3 ± 6.9 Hrz] linked with dyskinetic CBP in all patients. In total, 12 patients underwent TEM investigation, with 7 (58.3%) exhibiting a hallmark TEM defect. Furthermore, 17 patients underwent genetic testing and 7(41.2%) had a homozygous pathogenic variant. At the end, 12 (66.7%) patients had a confirmed PCD diagnosis, whereas 5 (29.4%) had a high-likelihood diagnosis. Table 2 shows the genetic variations and their pertinent related findings.

## 4. Discussion

To our knowledge, we present the largest number of Portuguese PCD patients who have been thoroughly investigated using HSVM, TEM, and genetic testing [13,18].

Because PCD is a complicated and genetically heterogeneous condition, confirming the diagnosis involves a multi-step process. Due to the nonspecific nature of the symptoms, although they are invariably present from early childhood, most patients remain undiagnosed until adulthood [3,13]. In our study, the mean PICADAR score was 6.2 points. While the PICADAR score can help to identify important clinical features, it is not an absolute diagnostic tool and should only be used for clinical screening.

A combination of clinical history, nNO, and HSVM findings is used to determine whether additional PCD study is necessary. However, even when one of the above parameters is normal, the investigation may proceed if the disease suspicion is strong, and the diagnosis of PCD is confirmed by TEM or genetics [19]. CBF and CBP should be included in the diagnostic work-up, according to ERS Guidelines; it is feasible to make a “highly likely” diagnosis when HSVM is consistently abnormal following culture, even with a normal TEM or genetics [1]. The detection of ciliary abnormalities by TEM is currently the most extensively used PCD diagnostic approach; nevertheless, approximately 30% of patients have a normal axonemal ultrastructure due to the TEM’s inability to identify some axoneme defects (*HYDIN*, *DNAH11*, and *GAS8*, for example) [1,20]. As a result, the use of HSVM to evaluate CBF and CBP allows some PCD diagnoses to be made until all genetic causes are identified.

The current investigation identified seventeen distinct variations, ten of which were classified as pathogenic and seven as questionable relevance. In the literature, half of PCD patients show *situs* abnormalities [21]; however, the frequency in our study was lower than expected (17.6%). Mutations in genes affecting central pair or radial spoke components (*RSPH1* and *RSPH4*), as well as in the genes involved in multiple cilia production, do not result in left/right body asymmetry [22]. In our *coorte*, the *DNAH5* mutation was found in two individuals with *situs inversus*. Unfortunately, due to the retrospective nature of the study, their HSVM and TEM data were not obtained. The third patient with *situs inversus* had the *CFAP298* mutation, which was linked to hyperkinetic CBP, as shown in other variants, such as *DNAH11* [23].

The clinical features associated with certain genotypes can vary greatly. When *RSPH1* is biallelic, it is associated with milder respiratory symptoms (lower prevalence of NRD, later onset of cough, and improved lung function) [5,21]. When we analyzed the 3 patients with this mutation, 2 had a history of NRD, and 1 had a forced expiratory volume in 1 s (FEV_1_) of 40% at the age of 44. Fortunately, none of them developed persistent colonizations.

Axoneme defects can result in male infertility due to sperm immotility or female subfertility due to deciliation found in Fallopian tubes. [2,13] In our study, patients with *CFAP298*, *DRC1*, *ODAD2*, *DNAAF3*, and *RSPH4A* mutations were infertile. However, the prevalence of infertility may be underestimated due to late parenting in today’s society.

In TEM, *RSPH4A* and *RSPH1* mutations are related with radial spoke apparatus absence and central pair abnormalities that are not independently diagnostic for PCD. Nonetheless, despite the fact that these alterations produce relatively mild phenotypes, the absence of those same structures in the sperm tail is sufficient to lead to infertility. In our population, the patient with the *RSPH4* mutation was infertile and the three sisters with the *RSPH1* mutation, despite not having performed fertility tests, were unable to conceive and chose to adopt [5].

## 5. Conclusions

Despite being a single-center retrospective study, it has the largest number of Portuguese PCD patients studied by the combination of HSVM, TEM, and/or genetic analysis. Most of our results are in agreement with the literature and demonstrate that PCD is underdiagnosed in Portugal, highlighting the need for greater awareness of the disease in daily clinical practice. When associated with other relevant symptoms, the history of infertility should be considered to support the PCD diagnostic hypothesis. More research is needed to improve PCD diagnostic test uniformity, including result interpretation.

## Figures and Tables

**Figure 1 genes-14-00541-f001:**
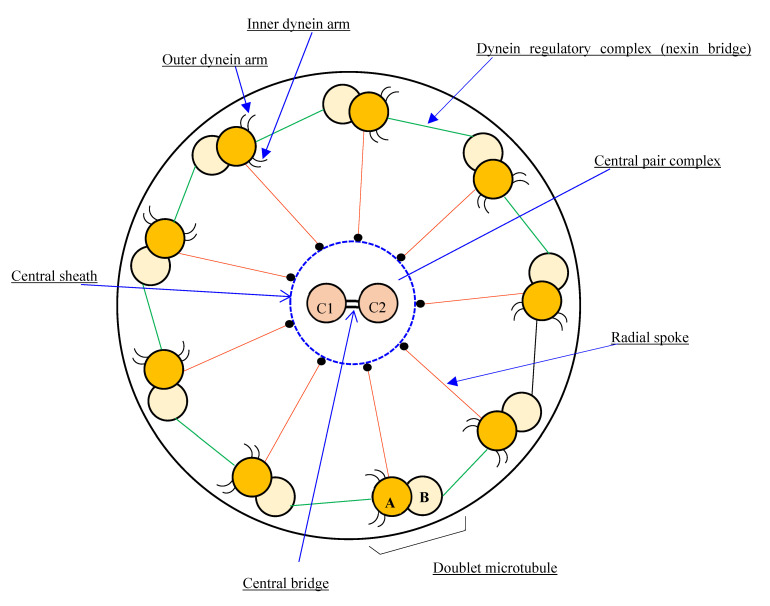
Diagram of the basic motile cilia structure showing the 9d + 2s microtubule arrangement pattern.

**Figure 2 genes-14-00541-f002:**
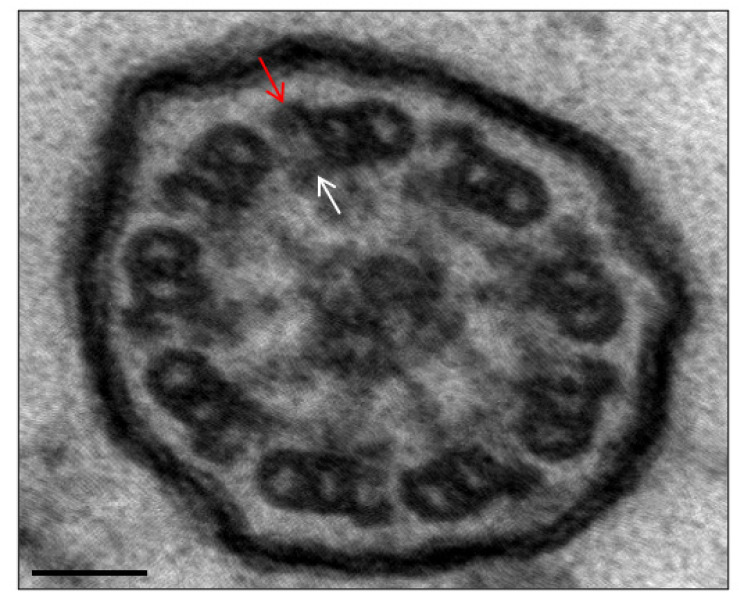
Ultrastructural image of a normal axoneme. Presence of the outer and inner dynein arms. Arrows point to outer (red) and inner (white) dynein arms as references. Bar: 50 nm.

**Figure 3 genes-14-00541-f003:**
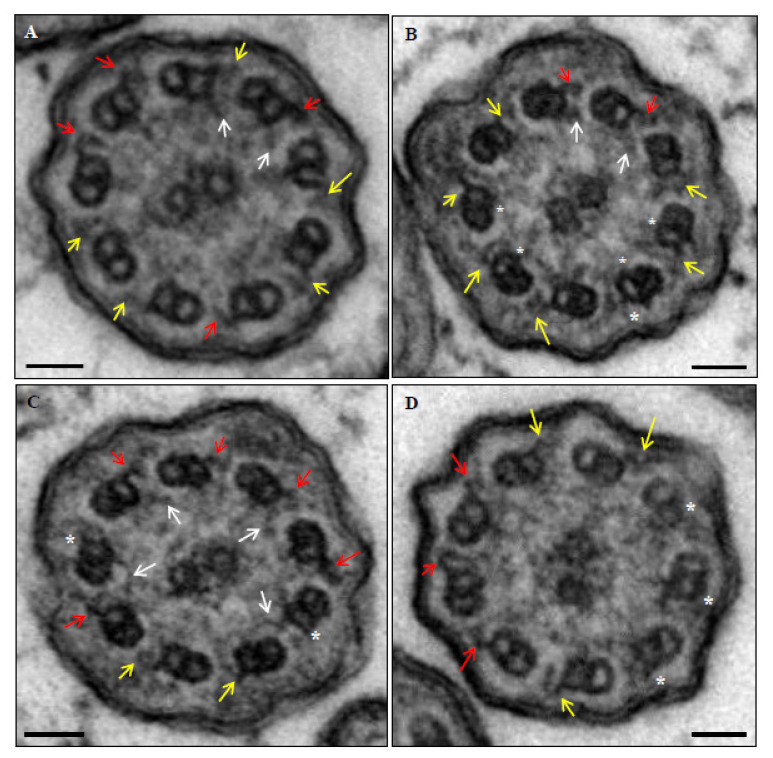
Ultrastructural images of axonemes from patient 6. Normal outer dynein arms (red), decreased dimensions or poor definition of outer dynein arms (yellow), normal inner dynein arms (white arrows), and absence of dynein arms (*). (**A**) Case with absent, decreased dimensions or poor definition of 5 outer and 7 inner dynein arms (abnormal axoneme). (**B**) Case with absent, decreased dimensions or poor definition of 7 outer and 4 inner dynein arms (abnormal axoneme). (**C**) Case with absent, decreased dimensions or poor definition of 4 outer and 5 inner dynein arms (normal axoneme). (**D**) Case with absent, decreased dimensions or poor definition of 6 outer and 9 inner dynein arms (abnormal axoneme). Bars: 50 nm.

**Figure 4 genes-14-00541-f004:**
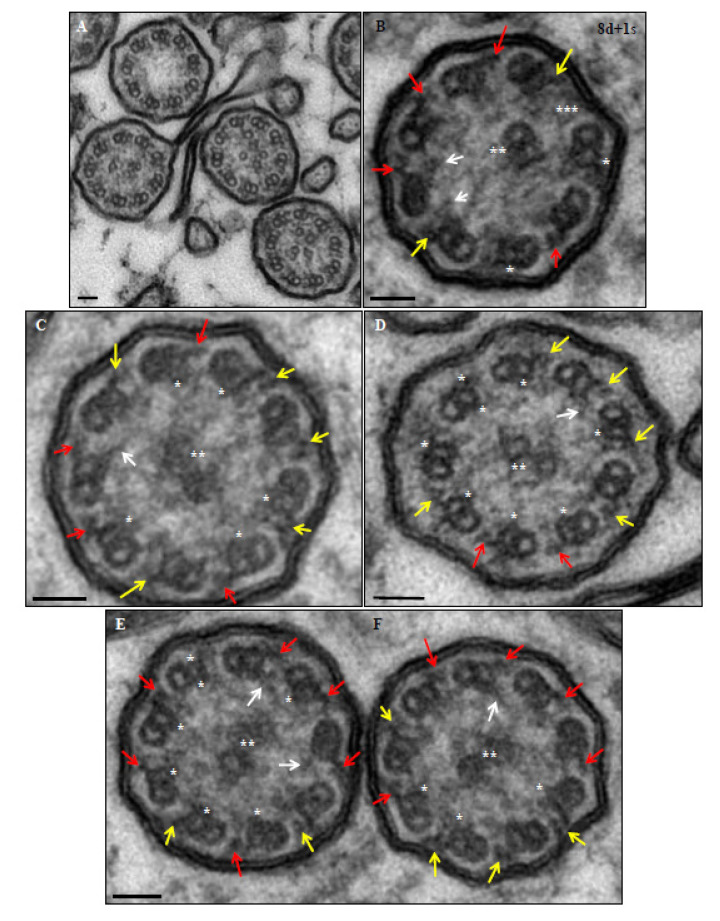
Ultrastructural images of axonemes from patient 15. Normal outer dynein arms (red); decreased dimensions or poor definition of outer dynein arms (yellow); normal inner dynein arms (white arrows); absence of dynein arms (*); absence, decreased dimensions or poor definition of the central pair microtubules (**); and absence of a peripheral microtubule doublet (***). (**A**) Axonemes with disruption of the central pair complex. (**B**) Case with absent, decreased dimensions or poor definition of 5 outer and 7 inner dynein arms (abnormal axoneme). (**C**) Case with absent, decreased dimensions or poor definition of 5 outer and 5 inner dynein arms (abnormal axoneme). (**D**) Case with absent, decreased dimensions or poor definition of 7 outer and 6 inner dynein arms (abnormal axoneme). (**E**) Case with absent, decreased dimensions or poor definition of 3 outer and 6 inner dynein arms (normal axoneme). (**F**) Case with absent, decreased dimensions or poor definition of 4 outer and 3 inner dynein arms (normal axoneme). Bars: 50 nm.

**Table 1 genes-14-00541-t001:** Characteristics of patients with primary ciliary dyskinesia at diagnosis. *n* = 17 (100.0).

Sex, female	12 (70.6)
Age, years	34.1 ± 11.3
BMI, kg/m^2^	23.7 ± 2.6
Smoking habits, never smoker	16 (94.1)
Clinical evaluation History of neonatal distress syndrome in term infants *Situs* abnormalities (dextrocardia and isomerism) Congenital cardiac defects Chronic productive cough Persistent rhinitis Chronic middle ear disease with or without hearing loss Infertility Positive family history Respiratory infections (previous year) Hospital admissions for respiratory infections (previous year)	8 (47.1) 3 (17.6) 0 (0.0) 16 (94.1) 14 (82.4) 10 (58.8) 6 (35.3) 4 (23.5) 1.9 ± 1.2 0.0 [0.0–0.0]
Scores PICADAR score FACED score BSI score	6.2 ± 2.3 1.0 [0.0–1.0] 4.2 ± 3.1)
Functional evaluation FVC (% of predicted) FEV_1_ (% of predicted) FEV_1_/FVC (% of predicted)	81.5 ± 13.1 73.5 ± 17.1 74.6 ± 9.5
Radiological evaluation Bronchiectasis Number of affected pulmonary lobes Inferior pulmonary lobes (left and/or right and/or medium)	17 (100.0) 3.0 [2.0–3.0] 12 (70.6)
Microbiologic evaluation Chronic colonization *Pseudomonas aeruginosa* *Haemophilus influenza*	3 (17.6) 2 (66.7) 1 (33.3)
FENO Nasal nitric oxide measurement (*n* = 4)	8.0 [5.0–10.0] 283.8 ± 266.5
HSVM Patients analyzed Ciliary beat frequency (Normal—12.75 Hz; interval 7.0–19.0) % of dyskinetic beat pattern	17 (100.0) 6.3 ± 6.9 86.1 ± 24.4
TEM Patients analyzed Outer dynein arm defects Outer dynein arm + inner dynein arm defects Inner dynein arm defects + microtubular disorganization Others	12 (70.6) 3 (25.0) 0 (0.0) 4 (33.3) 5 (41.7)
Genetic testing	16 (94.1)
Diagnosis Confirmed Highly likely	12 (70.6) 5 (29.4)

Data are presented as n (%), mean (± standard deviation) or median [range]; BMI: body mass index; FENO: fractional exhaled nitric oxide; FEV_1:_ forced expiratory volume in 1 s; FVC: forced expiratory vital capacity; HSVM: high-speed video microscopy; Hz: hertz; TEM: transmission electron microscopy.

**Table 2 genes-14-00541-t002:** Primary ciliary dyskinesia patients and their associated relevant diagnostic findings.

	Clinic at Diagnosis	PICADAR Score	HSVM (Normal: 12.75 Hz; Interval 7.0–19.0)	TEM (Hallmark Defects)	Gene	Variant	Status	Variant Classification	OMIM Phenotype/References
#1	♀, 51 years, *Situs inversus*, Infertility, FEV_1_ 70%, nNO 75,33 ppb, FENO 7 ppb	6	24.88 Hz, Hyperkinetic + RA	Absence ODA (Class 1)	*CFAP298*	c.524C > A(p.Ser175 *) c.302G > T(p.Gly101Val)	Heterozygosity Heterozygosity	Likely pathogenic Uncertain significance	615500
#2	♀, 20 years, FEV_1_ 80%, FENO 8 ppb	5	0.0 Hz, Immotile	*	*RSPH1*	c.275-2A > C	Homozygosity	Pathogenic	615481
#3	♀, 25 years, NRD, FEV_1_ 81%, FENO 8 ppb	6		*					
#4	♀, 44 years, NRD, FEV_1_ 40%, FENO 9 ppb	6		*					
#5	♀, 38 years, FEV_1_ 68%, FENO 24 ppb	8	8.72 Hz, RA + Ciliary stiffness	Normal ciliary ultrastructure	*DRC1*	c.352C > T(p.Gln118 *)	Homozygosity	Likely pathogenic	615291
#6	♀, 43 years, Infertility, NRD, FEV_1_ 70%, nNO 546 ppb, FENO 14 ppb	7	13.85 Hz, Coordinated and PI (31.3%) + Coordinated and RA (50.0%)	Absence IDA + MT disorganization (Class 1)					
#7	♂, 31 years, CC, FEV_1_ 62%, nNO 34 ppb, FENO 8 ppb	12	0.0 Hz, Immotile	Absence ODA and IDA + MT and CPC disorganization (Class 1)	*DNAAF11*	c.8G > A(p.Trp3 *)	Homozygosity	Pathogenic	614935
#8	♂, 8 years, *Situs inversus*, NRD, FEV_1_ 71%,	7	*	*	*DNAH5*	c.4237C > T(p.Gln1413 *) c.5290T > C(p.Ser1764Pro)	Heterozygosity Heterozygosity	Pathogenic Uncertain significance	608644
#9	♀, 26 years, *Situs inversus*, CC, FEV_1_ 90%	8	*	*		c.6962G > A(p.Trp2321 *) c.18552C > T(p.Arg618*)	Heterozygosity Heterozygosity	Pathogenic Pathogenic	
#10	♂, 30 years, NRD, FEV_1_ 80%, nNO 480 ppb, FENO 5 ppb	6	4.81 Hz, RA + Ciliary stiffness	Absence ODA (Class 1)		c.4237C > T(p.Gln1413 *) c.5157C > T(p.Phe1719 =)	Heterozygosity Heterozygosity	Pathogenic Uncertain significance	
#11	♂, 35 years, Infertility, NRD, FEV_1_ 98%, FENO <5 ppb	3	9.2 Hz, Coordinated and PI (11.8%) + Coordinated and RA (35.3%)	Absence IDA + MT disorganization (Class 2)	*ODAD2*	c.2408T > C(p.Val803Ala)	Heterozygosity	Uncertain significance	615451
#12	♂, 42 years, NRD, FEV_1_ 99%, FENO 23 ppb	2	7.5 Hz, Coordinated and PI (50.0%) + Coordinated and RA (31.3%)	Absence IDA (Normal)	*HYDIN*	c.5807C > T(p.Ser1936Leu) c.7837G > A(p.Ala2613Thr)	Heterozygosity Heterozygosity	Likely benign Uncertain significance	608647
#13	♀, 44 years, Infertility, FEV_1_ 72%, FENO <5 ppb	7	3.07 Hz, Coordinated and PI (35.7%) + Coordinated and RA (21.4%) + Immotile (35.7%)	No hallmark defects detected (Normal)	*DNAAF3*	c.1453G > A(p.Val485Met)	Heterozygosity	Uncertain significance	606763
#14	♀, 20 years, FEV_1_ 97%, FENO 5 ppb	8	4.62 Hz, Coordinated and PI (8.8%) + Coordinated and RA (44.1%) + Immotile (23.5%)	MT disorganization (Normal)	*DNAH1*	c.9211_9212del(p.Gln3071fs)	Heterozygosity	Pathogenic	617577
#15	♀, 42 years, Infertility, NRD, CC, FEV_1_ 66%, FENO 7 ppb	5	4.13 Hz, Coordinated and PI (14.3%) + Coordinated and RA (42.9%) + Immotile (28.6%)	MT and CPC disorganization (Class 2)	*RSPH4A*	c.961delT(p.Tyr321fs)	Homozygosity	Pathogenic	612649
#16	♀, 42 years, Infertility, CC, FEV_1_ 43%, FENO 11 ppb	6	7.67 Hz, Coordinated and PI (4.5%) + Coordinated and RA (45.5%) + Immotile (18.2%)	Absence IDA + MT disorganization (Class 1)	No mutation detected	-	-	-	-
#17	♀, 39 years, FEV_1_ 74%, FENO 10 ppb	3	0.0 Hz, Immotile (100%)	Absence IDA + MT disorganization (Class 1)	No mutation detected	-	-	-	-

CC: chronic colonization; CPC: central pair complex; FENO: fractional exhaled nitric oxide; FEV_1:_ forced expiratory volume in 1 s; HSVA: high-speed video microscopy; H: hertz; IDA: inner dynein arm; MT: microtubule; nNO: nasal nitric oxide; NRD: neonatal respiratory distress; ODA: outer dynein arm; OMIM: Online Mendelian Inheritance in Man; PCD: primary ciliary dyskinesia; PI: partial immotile; RA: reduced amplitude; TEM: transmission electron microscopy; Class: TEM classifications according to reference [9]. #__: Confirmed PCD diagnosis. * Test results not reported.

## Data Availability

Not applicable.

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
