# Peer review of "Primary Ciliary Dyskinesia in a Portuguese Bronchiectasis Outpatient Clinic"

_genes, 2023, doi:10.3390/genes14030541_

Round 1

Reviewer 1 Report

Review genes-2102463

1. Frequencies (n) should be presented without commas: n = 17 etc.

2. Standard deviations (SD) should be presented in parentheses: M (SD).

3. The better way to present syndromes and their proportions would be to show their combinations (comorbidity). Their sum must be equal to n and total proportion to 100%.  And it would be desirable to test the randomness of combinations.

Author Response

Frequencies (n) should be presented without commas: n = 17etc.

R: Corrected

Standard deviations (SD) should be presented in parentheses: (SD).

R: Corrected

The better way to present syndromes and their proportions would be to show their combinations (comorbidity). Their sum must be equal to and total proportion to 100%.  And it would be desirable to test the randomness of combinations.

R: Thank you for your comment. Regarding this point I decided to expose the symptoms individually because if it was express like a syndrome it would create many sub-groups and loose representability.

Reviewer 2 Report

The manuscript Primary Ciliary Dyskinesia in a Portuguese bronchiectasis outpatient clinic by Eduarda Tinoco * , Ana Rita Gigante , Edite Ferreira , Inês Sanches , Rute Pereira , Rosália Sá , Mário Sousa , Regina Monteiro , Ivone Pascoal  describes the features of Portuguese patients with PCD and analyses  their genetic variants.

It is obvious that authors made a great job and the results are signifiant.

The manuscript should definitely be accepted but after intensive corrections , the abstract, M&M, Results and Discussion should be rewritten .

English editing and proofreading is needed, please check spelling and typo

1.       Write an abstract section as a single paragraph, do not split in into Introduction,  M& M and results sections.

2.       Page 1, line 4 , typo ,  it should be airways

3.       Page 1? line 5, typo, should be defects

4.       Page 2, line 3 typo, example

5.       Page 2, line 3 typo next-generation ...

6.       Materials and methods section: please divide this section on subsections and describe more details about each method.

7.       Results: in M& M section you write about  TEM analysis, please provide a multipanel figure (if possible) of Axonemes and show the pathology.  It would definitely strengthen the MS, divide results into subsections in correspondence with  M&M section.

8.       Discussion: please compare your results with the same in the world practice, show similarities and the difference between populations and provide appropriate links.

9.       A schematic representation, comparing your results with literature data would also strengthen the manuscript.

10.   Check referencing style and reference list formatting.

Author Response

Write an abstract section as a single paragraph, do not split in into Introduction,  M& M and results sections.

R: Thank you for the recommendation; I agree, it is better.

Page 1, line 4 , typo,  it should be airways

R: Corrected

Page 1, line 5, typo, should be defects

R: Corrected

Page 2, line 3 typo, example

R: Corrected

Page 2, line 3 typo next-generation ...

R: Corrected

Materials and methods section: please divide this section on subsections and describe more details about each method.

R: Thank you for your suggestion. I agree with it and already made some alterations and included more information regarding each method.

Results: in M& M section you write about  TEM analysis, please provide a multipanel figure (if possible) of Axonemes and show the pathology.  It would definitely strengthen the MS, divide results into subsections in correspondence with  M&M section.

R: Thank you for the suggestion. It was possible to provide some multipanel figures of axonemes.I also wrote a paragraph regarding the pathology.

Discussion: please compare your results with the same in the world practice, show similarities and the difference between populations and provide appropriate links.

R: I improved my discussion taking in mind your suggestions.

A schematic representation, comparing your results with literature data would also strengthen the manuscript.

R: Thank you for your comment. As far as I can tell there is only two national studies regarding PCD that account for 13 pediatric patients. As it was a very small coorte of patients we choose not to present a schematic representation, comparing this results.

Check referencing style and reference list formatting.

R: Checked

Reviewer 3 Report

Your research was adequately addressed.

Author Response

Thank you very much

Reviewer 4 Report

The manuscript reports clinical features and genestics of patients with PCD, a rare disorder. The paper has the credit to add a dataset of PCD patients to the literature. 

Major limitations are: 1) the low number of patients: 2) the descriptive nature of the manuscript. 

My suggestions: 

1) please verify if it is possible to correlate some clinical parameter ( as an example age of onset of symptoms) with genetics ( truncating mutation?); 

2) Please provide more info about genetic analysis in the method section, THE methods used for variants prioritization; 

3) Please provide the explanation of PICARD score, before using it in the results;

4) Please expand the discussion section and re-modulate in more clear and focused subparagraphs. The following issues should be covered: 1) difficulties of diagnosis; 2) genotype to phenotype correlation, if any ( discuss why there are limited data on this); 3) comments on the data provided in this study, highlighting novelties.

Author Response

Please verify if it is possible to correlate some clinical parameter (as an example age of onset of symptoms) with genetics (truncating mutation?); 

R: Thank you for your suggestion. I extended my research traying to find some correlations.

Please provide more info about genetic analysis in the method section, the methods used for variants prioritization; 

R: Thank you for your suggestion. I agree with it and already made some alterations and included more information regarding genetic analysis.

Please provide the explanation of PICADAR score, before using it in the results;

R: Thank you for the recommendation, I provided the explanation;

Please expand the discussion section and re-modulate in more clear and focused subparagraphs. The following issues should be covered: 1) difficulties of diagnosis; 2) genotype to phenotype correlation, if any (discuss why there are limited data on this); 3) comments on the data provided in this study, highlighting novelties.

R: I improved my discussion taking in mind your suggestions.

Round 2

Reviewer 2 Report

Dear authors, you've done great work! I' m satisfied. I read the revised version of manuscript with a great interest.  However, I have several minor comments:

1) please, for each Figure provide a short title

2) on figures 2-4 provide a short self explanatory capture

3) delete A from figure 2, there is only one image

4) on figures 2-4 add a scale bar

5) go through the text and check typos

Author Response

1) Please, for each Figure provide a short title

R: Done

2) On figures 2-4 provide a short self explanatory capture

R: It was difficult for us to understand what was asked… Hope we have addressed this topic

3) Delete A from figure 2, there is only one image

R: Done

4) On figures 2-4 add a scale bar

R: Done

5) Go through the text and check typos

R: Checked

Reviewer 4 Report

The manuscript provides an interesting dataset of well characterized patients to the literature.

Given the rarity of the condition, even if the number of patients is not very high it deserves attention.  

Author Response

Thank you very much for your comment